# A Novel Design towards Reducing Leakage Loss for Variable Geometry Turbines

**Kai Zhou** [1,*] **and Xinqian Zheng** [2]

1 Institute for Aero Engine, Tsinghua University, Beijing 100084, China
2 Turbomachinery Laboratory, State Key Laboratory of Automotive Safety and Energy, Tsinghua University, Beijing 100084, China
* Correspondence: kinozhou@tsinghua.edu.cn

**Abstract:** To accommodate the next generation of adaptive/variable cycle engines and gas turbine power, the variable geometry turbine (VTG) is widely acknowledged as a most essential component. VGT consists of an adjustable vane to address the combined goals of high dry thrust and low specific fuel consumption (SFC) at subsonic cruises for aero-engines. This concept allows an engine to operate at a constant bypass ratio over a wide range of pressure ratios. To avoid scraping during rotation for guide vanes, a typical gap is deliberately left, which leads to significant leakage loss. In this research work, a novel spherical convex plat with a pivot shaft is proposed, which can be obtained by additive manufacturing. The plat is sophisticatedly designed according to the aggressive tip/hub pressure gradient. This design naturally generates a blockage for the gas from the pressure side towards the suction side. As a result, the most aggressive pressure gradient is removed, and maximum leakage flow is eliminated. The overall leakage loss is reduced. This simple rotating structure design can improve the efficiency by 0.4–3.0% within the wide range considered. Based on the understanding of the loss mechanism, a radially restacked vane is designed and another extra 0.2% improvement is achieved. This universal design philosophy is also verified on different loading blade profiles, i.e., front-, middle- and aft-loaded turbine vane. The improved aerodynamic performance is achieved with this novel idea.

**Keywords:** variable geometry turbine; leakage loss; low-pressure turbine

## 1. Introduction

To accommodate the next generation of adaptive/variable cycle engines and gas turbine power, the variable geometry turbine (VGT), also known as variable area turbine nozzle (VTN), is widely acknowledged as the most essential component. VGT addresses the combined goals of high dry thrust and low specific fuel consumption (SFC) at subsonic cruise for aero-engines (Keith et al. [1]). For multiple working conditions in marine intercooled and regenerative (ICR) gas turbines, the variable geometry turbine is also required to match the turbine and compressor in the wide range of mass flow rate and rotation speed. Variable geometry turbines are also widely found in diesel engines to meet the requirements of the stringent emission standards (Huang et al. [2]).

To reach this goal, a throat-area adjustable structure is usually adopted in VGT design to accommodate the wide-range mass flow and pressure ratio. The ability of VGT in part-loaded performance is treated as the most significant improvement (Haglind [3]). The access to implement such a structure involves an important consideration: it requires a simple and robust mechanical design and also requires a high efficiency for wide-range conditions. This multi-objective turbine design is very important for turbomachinery engineers. Fang et al. [4] used a harmonic design method to obtain a new performance in a wide operating range. The same research group [5] also investigated the characteristics of two typical modes, i.e., the low bypass ratio and the high bypass ratio in the unsteady

environment. There is a 2.5% efficiency gap between the two modes and evident unsteady pressure fluctuation is observed. Many other open publications can be found dealing with the simple geometry without any tip/hub surface modification, which cannot be neglected.

Compared to the ideal situation in VGT, from the perspective of the aerodynamic design of adjustable turbine vanes, the most significant challenge encountered is the leakage loss, both from the hub region and tip region. These gaps are deliberately left to avoid heat expansion or scraping due to rotation. Evident clearance leakage flow forms due to the pressure difference between the pressure side and the suction side across the tip gap of the vane. The leakage flow ejects into the suction side passage and interacts with the main flow, causing flow blockage and mixing loss, similar to the situation in the rotor tip leakage flow (Tallman and Lakshinarayana [6,7]). As argued in Schaub et al. [8], for a low aspect ratio turbine, the tip leakage flow could account for up to one-third of the aerodynamic loss in a turbine stage. A similar conclusion is found in the VGT design. Niu et al. [9] experimentally investigated the leakage loss and found that not only does the rotating angle play an important role but so does the loss changes with exit Mach number. Gao and Huo [10] performed a detailed analysis on tip clearance with a rotating bar and found a similar trend.

To eliminate the clearance leakage loss, many methods and patents can be found. Yue et al. [11] considered a spherical endwall, as well as nonuniform clearance. The leakage flow interacts with the passage vortex, thus inducing additional loss. A spherical endwall with a uniform clearance significantly improves the efficiency as large as 0.8%. Many other attempts deal with how to eliminate the loss, such as a winglet design (Gao et al. [12]), 3D contoured vane endwall [13], part blade rotation [14], a unique cam-driven clamshell high-pressure turbine [1], floating sealing [15] and many others. However, most of the design is either at the cost of the very complex structure to realize [12,14–16], or the structure is not suitable in the extremely hot gas environment [12,14,15]. This work is involved in proposing a simple geometry configuration to eliminate the leakage loss. The design philosophy understands the basic loss mechanism and helps to enlighten the designers.

In this research work, a novel spherical convex plat with a pivot shaft is proposed and applied to the variable geometry low-pressure turbine vane. The installed position is deliberately chosen to coincide with the maximum leakage region. By the application of the pivoting shaft and the spherical convex plat, part of leakage mass is blocked, thus reducing the leakage loss. To verify the design philosophy, a radial restacked vane is redesigned, and an extra 0.2% efficiency improvement is achieved. Furthermore, three typical loading blades are analyzed to underline the design philosophy: front-, middle-, and aft-loaded blades. All of them show a similar trend in improving the efficiency by 0.4–0.65%, which proves the superiority of this novel spherical convex design.

## 2. Methodology

A typical low-pressure turbine from a marine gas turbine was chosen for the current study (Yue et al. [11]). A total pressure condition was imposed on the inlet boundary condition, and a static pressure condition (standard atmospheric pressure) was placed on the exit boundary condition. The overall total-to-static pressure ratio was 1.33. The inlet total temperature was set to be 1051 K. The design speed was 3270 rpm. The main parameters of the LPT are listed below in Table 1. For the consideration of leakage effect, a stator tip and hub clearance were deliberately left when adjusting the vane stagger angles. Here, a typical value of 1.0 mm clearance, i.e., 1.1% $C_x$, was set at the design stagger angle. The wide-range performance was investigated at about $\pm 10$ degrees around the design angle.

**Table 1.** Parameters of the low-pressure turbine.

| Stator | No. | 47 |
|---|---|---|
| Rotor | No. | 55 |
| Stator Axial Chord, $C_x$ | mm | 87.5 |
| Span, h | mm | 150.0 |
| Stator Tip/Hub Clearance, $\tau/C_x$ | / | 1.1% |
| Rotor Tip Clearance, $\tau/C_x$ | / | 1.1% |
| Pivot Shaft Radius, $R/C_x$ | / | 5.5% |
| $r_t/C_x$, $r_h/C_x$ | / | 57.1% |
| $d_t/C_x$, $d_h/C_x$ | / | 2.2% |

*2.1. Numerical Settings*

Numerical methods were adopted in this study. The overall computational domain consisted of a stator domain and a rotor domain, as shown in Figure 1. A robust tetrahedron grid was generated by ANSYS FLUENT Meshing. Figure 2 shows the mesh details within the 'boundary layer' near the leading edge and within the tip gap. Triangle prisms were applied to the wall surface. The very first layer of boundary surface y+ was carefully controlled to be 1–5. A total of 25 mesh layers were spread within the 'boundary layer' near the wall, and the mesh growing ratio was 1.3. Commercial solver CFX was used to solve the discrete Reynolds-Averaged Navier–Stokes (RANS) equations. A mixing-plane model was applied to the stator–rotor interface. In total, only two passages (one stator passage and one rotor passage) were needed in the simulations. The second-order central difference scheme for the space was used to discretize the equations for the mass flow, turbulent kinetic energy and the specific dissipation rate. Based on previous experiences [17,18], shear stress transport $k - \omega$ turbulence models (Menter [19] and Menter et al. [20]) were chosen for the LPT study as the turbulence models have an advantage in terms of the capability of capturing the detailed tip leakage flows and the mixing loss. The case was treated to be convergent only if the residue of mass flow rate was below $10^{-4}$. The convergence level of other parameters, such as velocity and turbulence kinetic energy, was of a similar order.

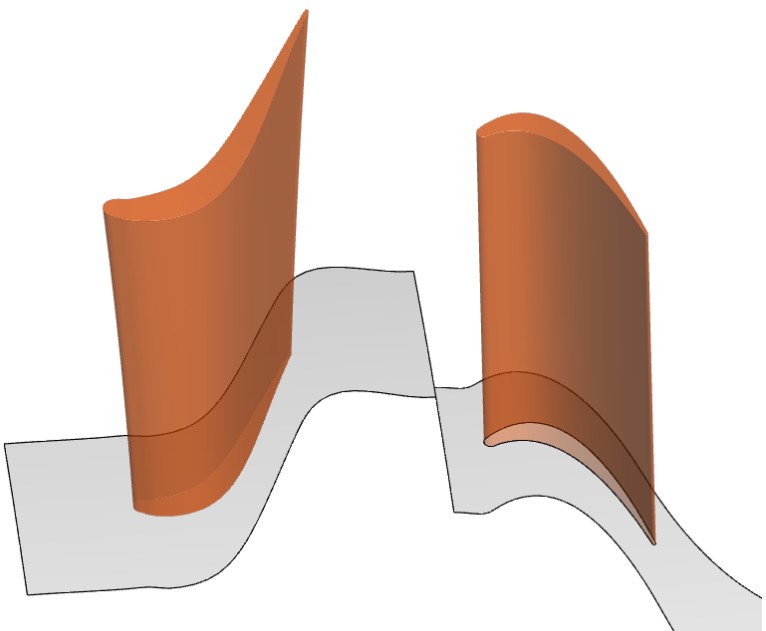

**Figure 1.** A typical low-pressure vgt from a marine gas turbine.

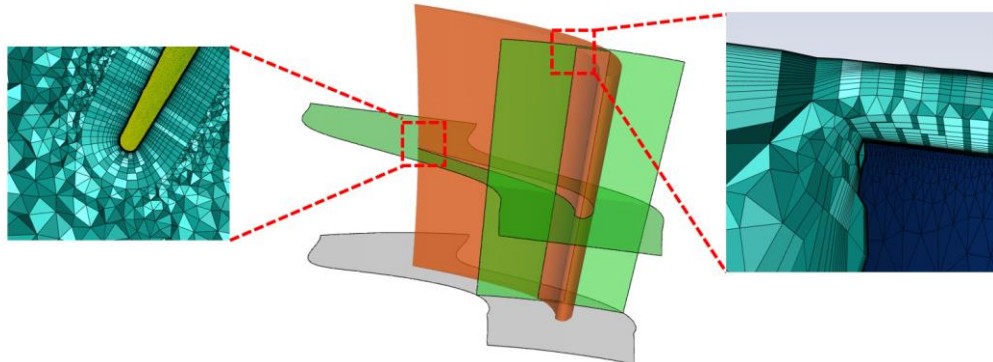

**Figure 2.** Mesh details near the vane trailing edge and tip gap.

### 2.2. Mesh Independence

The domain was filled with a tetrahedron grid. To access the credibility of numerical simulations, a mesh independence study was conducted by varying the mesh nodes in the stator domain from $2.0 \times 10^6$ to $8.0 \times 10^6$ while keeping the constant node number in the rotor domain. The overall efficiency changed from 90.65% to 91.05%. After the mesh nodes were over 6 million, the efficiency variation was within 0.03%, as shown in Figure 3. In this situation, a fine mesh with $6.1 \times 10^6$ nodes, or $2.4 \times 10^6$ cells, was sufficient to underline the efficiency evaluation. Similar nodes number was dispersed within the rotor domain. Overall nodes for a simulation case were $1.05 \times 10^7$.

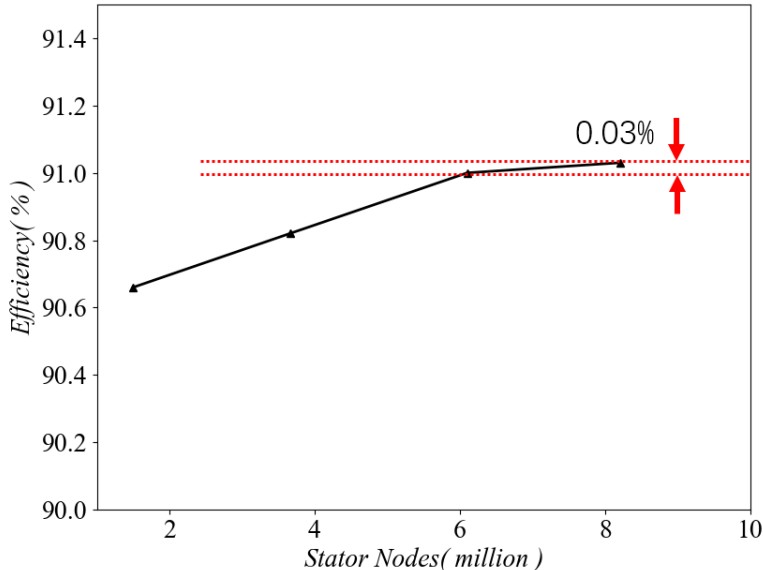

**Figure 3.** Mesh independence with stator nodes numbers.

### 2.3. A Novel Spherical Convex Plat

The case with a stator tip and hub gap was taken as a basic datum. What should be noted is that, as the hub and shroud surface was a ring surface, the clearance along the tip/hub blade surface was non-uniform after rotation. The effect of non-uniform clearance was very limited (not shown here) in the current study as the blade profile was moderately loaded and most of the loading was concentrated near the pivot point. As a result, the non-uniform clearance effect was neglected in the current research.

Based on the datum case, a novel spherical convex plat was imposed on the tip and hub region, as shown in Figure 4. A pivot shaft was installed through the blade. The design philosophy will be given in Section 3. The main geometric parameters are shown in Table 1. The spherical radius near the tip and the hub was chosen to be the same, $r_t / C_x = r_h / C_x = 57.1\%$. The

plat was slightly higher than the tip surface about 2.2% $C_x$, forming a convex plat. The diameter of the rotating pivot shaft 2R was 11.0% $C_x$, which is close to the local blade thickness. It was also chosen for the sake of structural stability. In the later part, the datum case with clearance and the new design were named 'Datum' and 'Convex', respectively.

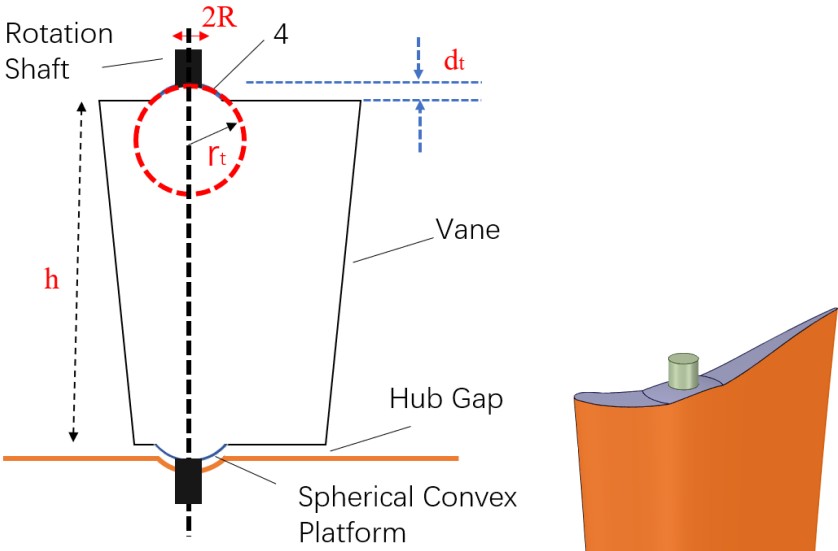

**Figure 4.** A novel spherical convex plat with rotating pivot shaft.

## 3. Results and Discussion

### 3.1. Design Philosophy

The spherical convex plat is built based on the pressure distribution of the datum case at the design angle. The static pressure coefficient, which is defined in Equation (1), is presented in Figure 5. The datum case presents the most aggressive pressure gradient driven across the gap, as circled by 'A', where the local leakage flow contributes a significant amount of the overall leakage mass flow rate. In the Convex case, the spherical convex plat with pivot shaft coincides with the region 'A'. As a result, a large amount of mass flow is blocked by the convex plat and the shaft. The most aggressive pressure gradient is removed, as marked as 'B'. A similar trend is seen in region 'C'. The reason why a spherical convex plat is imposed is that it causes a small blockage, as well as acting as insurance to keep the clearance constant before and after the vane rotation.

$$C_p = \frac{P_{0,in} - p}{P_{0,in} - p_{ex}} \tag{1}$$

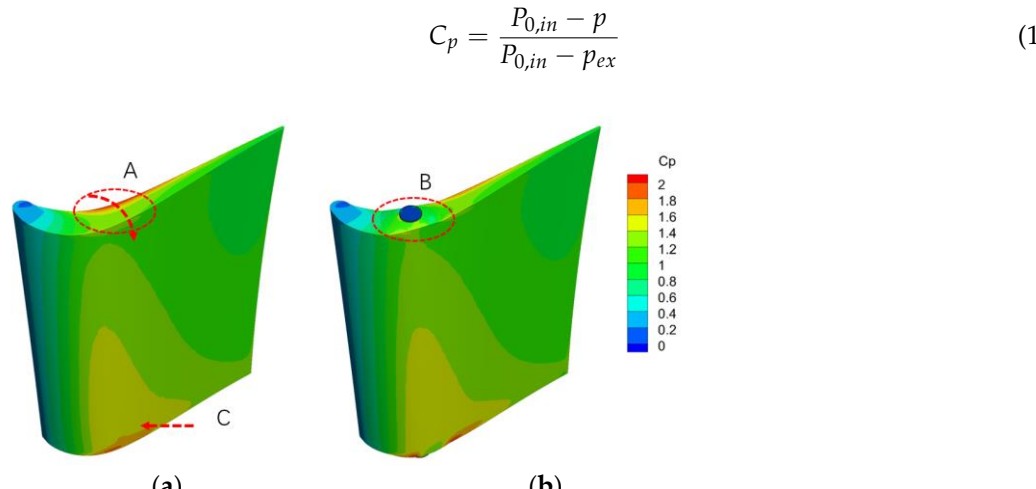

**Figure 5.** The cp distribution on the vane surface at design angle. (**a**) Datum (**b**) Convex.

### 3.2. Flow Field Analysis

The resulting streamlines near the tip region are also compared in Figure 6, where the streamlines are colored by velocity magnitude. As mentioned above, when the flow is driven across the tip gap, the highest velocity magnitude is identified above the region of the most aggressive pressure gradient. In the Convex case, when the flow impinges onto the pivot shaft, the flow streamlines are stopped and passed around the circular obstruction. As a result, part of the flow is disturbed and a low-momentum region exists behind the shaft, marked as 'D'. The corresponding stagnation pressure coefficient on the vane exit surface is compared in Figure 7. In the datum case, the high total pressure coefficient is related to high loss, which is contributed to by the secondary flow, including the tip leakage flow and the accompanied passage vortex. As shown in Figure 7a, the hub loss is more significant than the loss near the tip because the hub blade region is designed to have higher loading. The convex design limits the loss near the casing, as well as the loss near the hub. What should be noted is that the loss reduction near the casing 'E' is much more significant than that near the hub region 'F'. This is because the blade profile is straightly stacked to have a higher loading towards the hub. However, the convex plat position is chosen based on the pressure distribution on the tip surface, which leads to a small deviation between the plat center and the maximum leakage region on the hub region. As a result, the benefit due to the blockage is less identified than that near the casing. This indicates that a sophisticatedly designed radially stacked blade, combined with the current convex plat design philosophy, is a good option in the VGT design. The restacked profile is presented in Section 3.4, and more improvement will be achieved by this novel design.

$$C_{P0} = \frac{P_{0,in} - P_0}{P_{0,in} - p_{ex}} \tag{2}$$

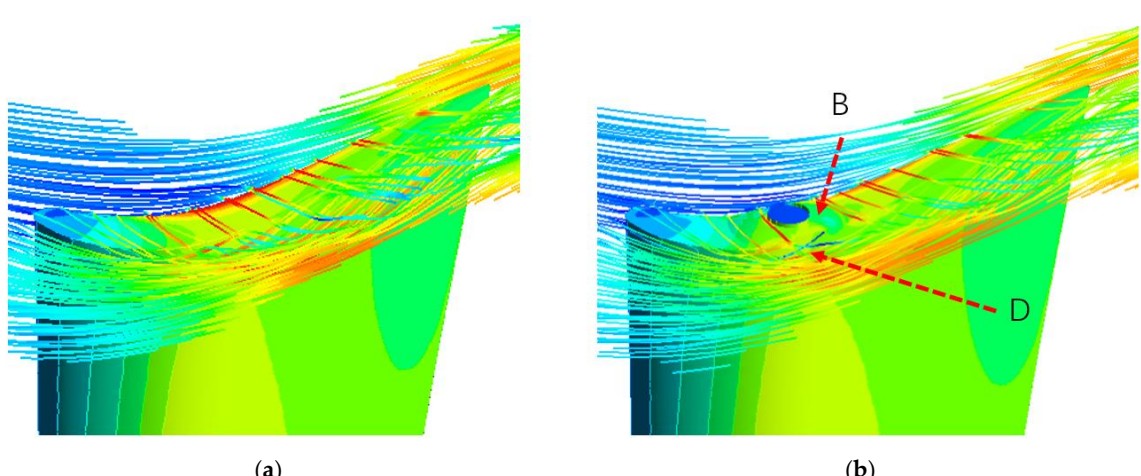

|  (a)  |  (b)  |

**Figure 6.** Streamlines near the tip region at design angle. (**a**) Datum (**b**) Convex.

### 3.3. Working Characteristic Lines

According to the basic target for VGT, the vane is required to operate between $-10$ degrees and $+10$ degrees around the design stagger angle (50 degrees) to achieve the realization of the wide range of throat-area variation. The mass flow rate is non-dimensionalized by the mass flow at the designed stagger angle. In this case, the mass flow rate changes almost linearly with the stagger angle increases, as shown in Figure 8, regardless of the Datum case and the Convex case. For every degree of variation, there is a corresponding 3% mass flow rate changing.

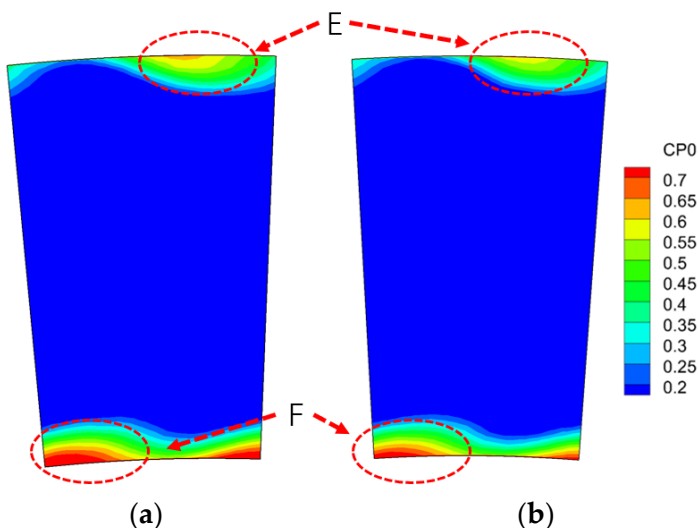

**Figure 7.** Stagnation pressure coefficient on the vane exit surface. (**a**) Datum (**b**) Convex.

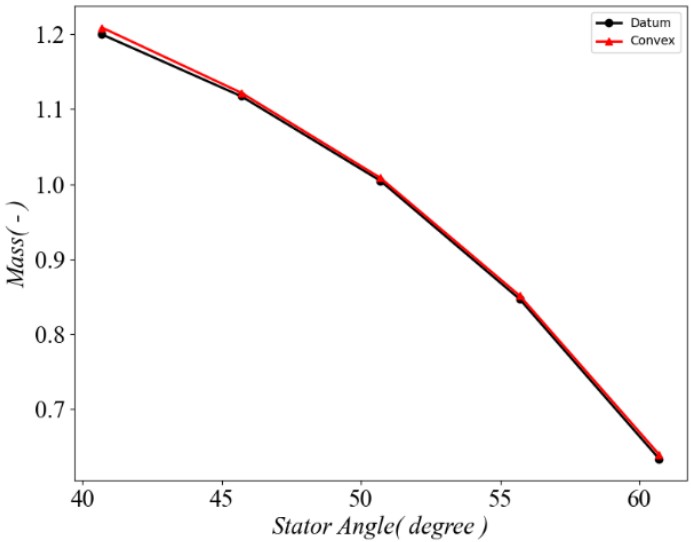

**Figure 8.** Mass flow rate versus the angle, at designed speed and pr.

A 100% speed line and 120% speed line are presented in Figure 9. With the designed pressure ratio considered at 100% speed, the overall efficiency increases slowly with the stagger angle, reaches the peak efficiency, and then drops quickly after the designed stagger angle (around 50 degrees). The peak efficiency occurs at the designed stagger angle (around 50 degrees). The sharp drop is the result of velocity triangle match. When the stagger angle increases over 50 degrees, the relative attack angle for the rotor blade is quite large, leading to a huge loading for the rotor blade profile, as well as the separation, which is corresponding to the sharp efficiency drop. At a higher speed of 120%, the datum case shows an evident efficiency drop after 55 degrees and a much sharper decline towards the higher angle. After the novel convex structure is imposed on the vane, a relatively small increase (around 0.4–1.2%) is achieved at 100% speed. However, at a higher speed of 120%, a much larger improvement, which is up to 0.6–3.0%, is achieved, especially at a higher stagger angle. At this speed, peak efficiency is around 55 degrees. The huge improvement is related to the velocity triangle and vane loading, where the convex blockage is more effective on the higher loaded blades.

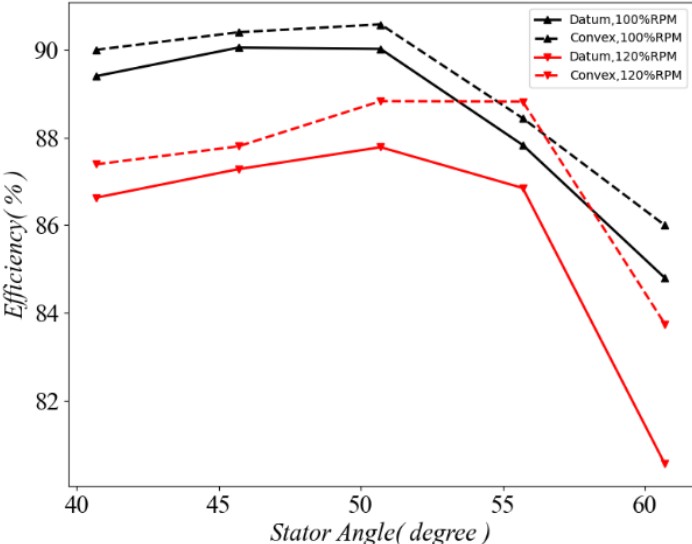

**Figure 9.** Efficiency versus the angle with different rotation speed, at designed pressure ratio.

The different characteristics of the low-pressure turbine under different pressure ratios at the designed speed are shown in Figure 10, where 0 degrees means the designed stagger angle and ±5 degrees represents a bias from the designed angle. Within the pressure ratio range from 1.33 to 2.5, the turbine efficiency at the design stagger angle almost linearly decreases with the pressure ratio, as this turbine is specially designed for a low-pressure ratio environment. At a smaller stagger angle, the turbine remains highly efficient with the pressure ratio. At a higher stagger angle, however, the efficiency drops quickly with the pressure ratio. This indicates that in designing VGT, when the turbine operates at a higher stagger angle, engineers should carefully choose the co-working lines as the turbine works efficiently only within a very narrow range of pressure ratio. After the application of the novel convex plat, an evident improvement is achieved, with similar distributions along with the pressure ratios.

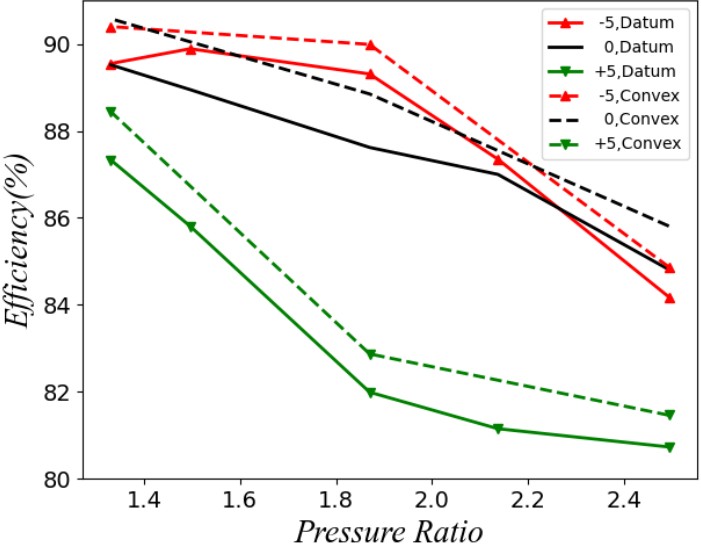

**Figure 10.** Efficiency versus pressure ratio, at the designed speed.

### 3.4. A Restacked Profile Design

As shown previously in Figure 7, the hub secondary flow contributes more loss than the tip leakage flow. However, the loss reduction near the hub is not as significant as that near the tip. This is because the convex plat does NOT exactly coincide with the maximum

hub leakage region, i.e., the most aggressive pressure gradient. Figure 11 shows the static pressure coefficient on the hub surface. The maximum leakage flow passes along the purple dashed arrow. However, the convex plat is located upstream of the arrow, which leads to an unsatisfactory improvement. This original vane profile is not specially designed with the consideration of the convex plat philosophy.

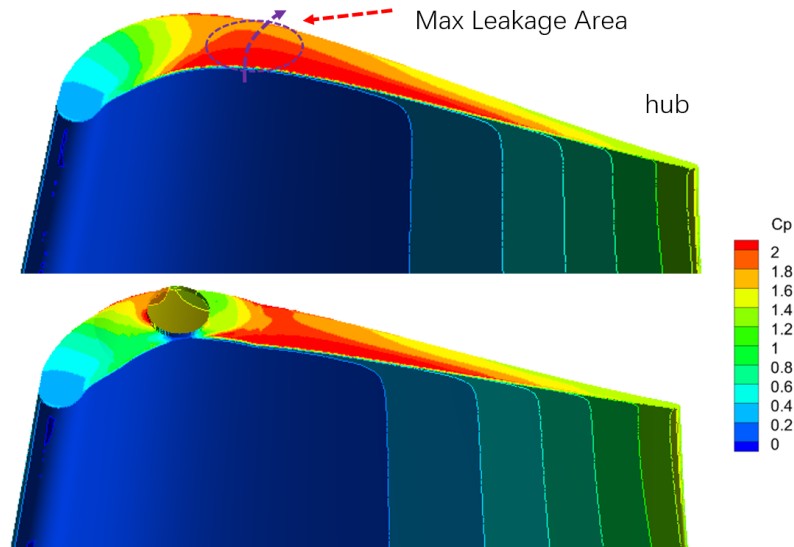

**Figure 11.** Static pressure coefficient on the hub surface; upper: datum hub, lower: convex hub.

To further improve the turbine efficiency, as well as to underline the novel design philosophy, a sophisticatedly restacked vane is redesigned, as shown in Figure 12. Compared to the Convex case, the hub profile is moved towards the leading edge slightly so that the maximum leakage location on the hub surface is radially overlapped with the maximum leakage location on the tip surface, whose design is named a 'Restacked' case. With this design, another 0.2% improvement is achieved. As a matter of fact, the radius of the pivot shaft is only 90% of that in the Convex case, as the local thickness is restricted by the hub blade profile. This means the blockage effect due to the pivot shaft is reduced in the Restacked design. However, by eliminating the hub loss, which dominates the overall loss within the vane, a better design could be achieved. The stagnation pressure coefficient near the hub is compared in Figure 13. The peak loss marked as 'F' is suppressed, as well as the high-loss area. This also highlights Denton's basic principle in turbomachinery: the ability to optimize the turbomachine is highly related to the knowledge of flow physics [21].

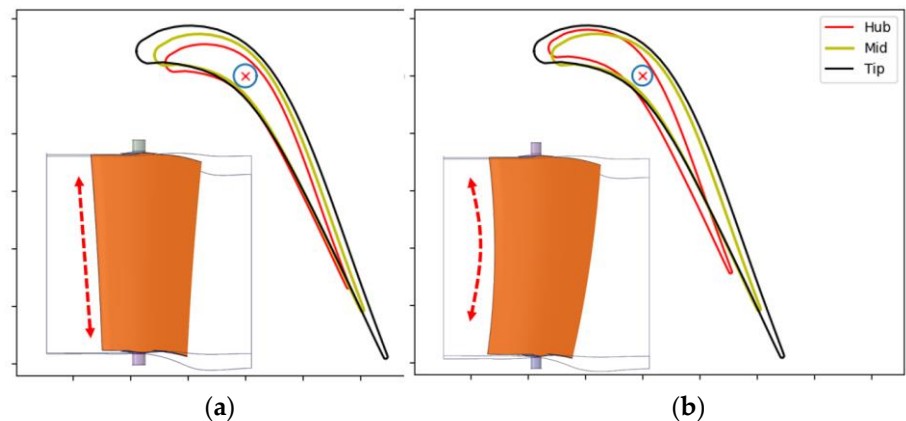

**Figure 12.** Radially restacked vanes, before and after. (**a**) Convex (**b**) Restacked.

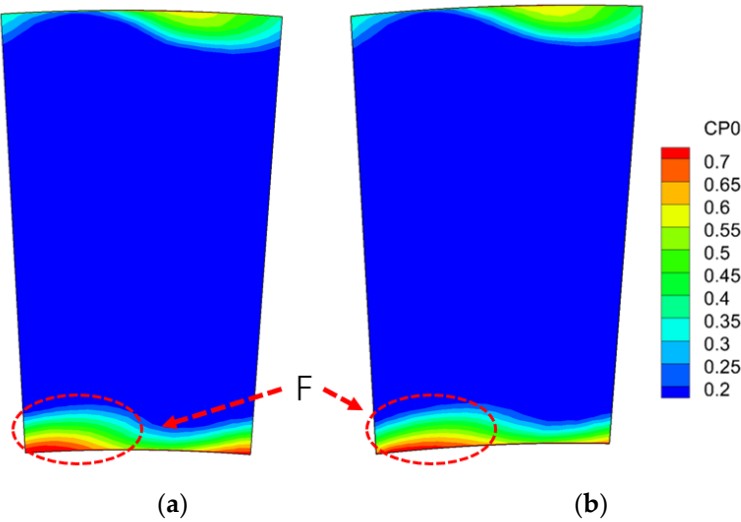

**Figure 13.** Stagnation pressure coefficient on the vane exit. (**a**) Convex (**b**) Restacked.

## 4. Load Effect

According to the argument above, the spherical convex plat is designed based on the datum pressure distribution. To explore the effect of loading distribution on the loss reduction with this novel structure design, three typical cases are artificially investigated, i.e., the front-, the middle-, and the aft-loaded blades. To simplify the study, those three vanes are deliberately made with straight blades, which are shown in Figure 14. These three types are normalized by the axial chord, respectively. The blade loading (Zweifel number) is kept the same as the blade is designed to be the same inlet and exit flow angle. The middle-loaded profile is taken from the middle span profile in the Datum case. So, the radial exit flow angle is different from the Datum case in Section 3. However, this part is just to clarify the effect of loading distribution with this novel design. Therefore, the spanwise loading effect is neglected. Among three cases, the stage loading and reaction keep exactly the same.

$$Ma_{isen} = \sqrt{\frac{2}{\gamma - 1}[(\frac{P_{0in}}{p})^{\frac{\gamma - 1}{\gamma}} - 1]} \tag{3}$$

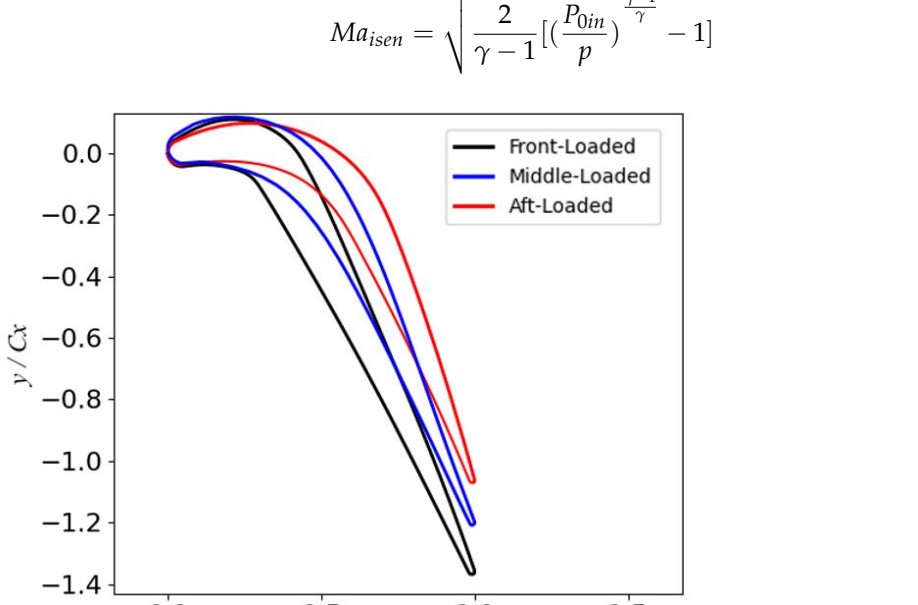

**Figure 14.** Vane profiles with front-, middle-, and aft-loaded distribution.

The static pressure coefficient on the midspan surface is presented in Figure 15. It shows that the pressure distribution is only affected near the maximum loading area, as circled in red. The rest of the pressure distribution is almost the same. The corresponding loading along the blade surface is given in Figure 16, where the blade loading is presented by the isentropic Mach number, which is defined in Equation (3). In the research, the blade loading is artificially generated and the reverse design is conducted by MISES 2.53 to obtain the blade profile in Figure 14. The front-loaded profile has a significant peak value of isentropic Mach number around $0.4C_x$, marked as 'A', while the middle-loaded vane and the aft-loaded vane have a peak loading around $0.5C_x$ and $0.65C_x$, respectively. According to the convex plat design philosophy, three uniform clearance cases are calculated first. After the most aggressive pressure gradient region is identified, a spherical convex plat with a rotating pivot shaft is imposed around the targeted region. The two typical geometries and corresponding $C_p$ are presented in Figure 17. In Figure 17a, the peak loading on the suction side is more concentrated, while in Figure 17b, the peak loading is less significant. The convex plat, along with the pivot shaft, is rightly located at the maximum $C_p$, both for the front-loaded vane and the aft-loaded vane.

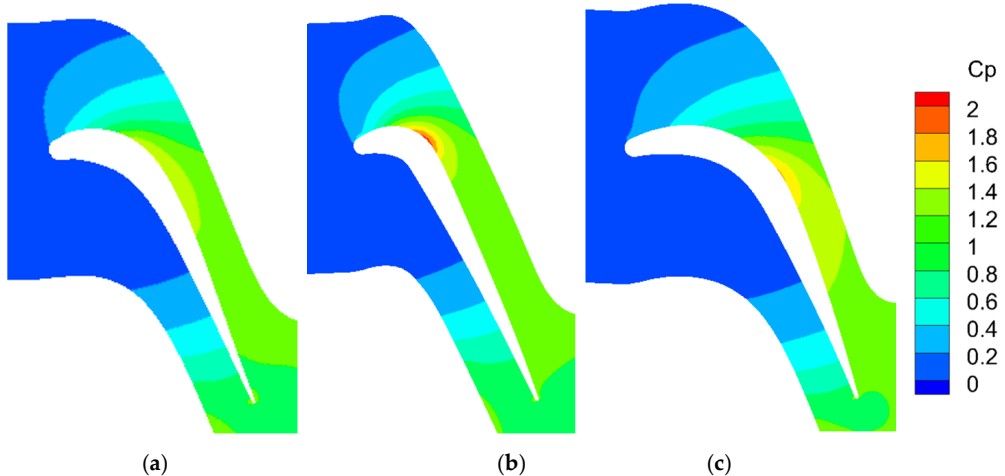

     (**a**)                   (**b**)              (**c**)

**Figure 15.** Static pressure distribution on 50% span of stator domain. (**a**) Middle-loaded (**b**) Front-loaded (**c**) Aft-loaded.

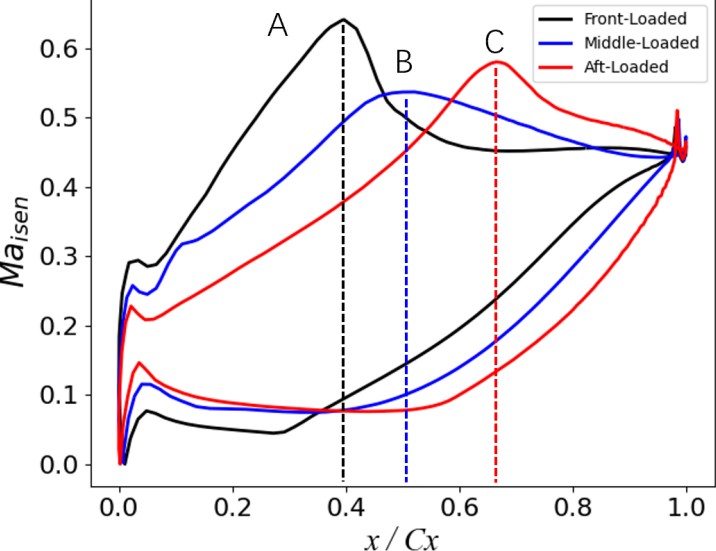

**Figure 16.** Different loading distribution on three vanes.

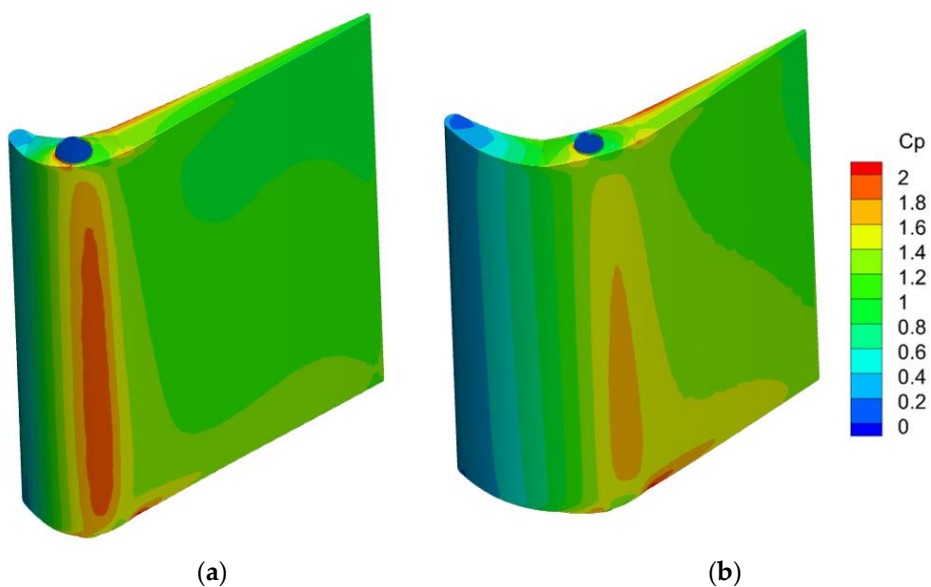

**Figure 17.** The cp on two typical vane surfaces, at design speed and design pressure ratio. (**a**) Front-Loaded (**b**) Aft-Loaded.

The improved efficiency (efficiency difference between the uniform clearance case and the convex case, respectively) is exhibited in Figure 18. A more beneficial efficiency gain is achieved in the front-loaded case. The middle-loaded case ranks second, and the aft-loaded case shows the smallest gain.

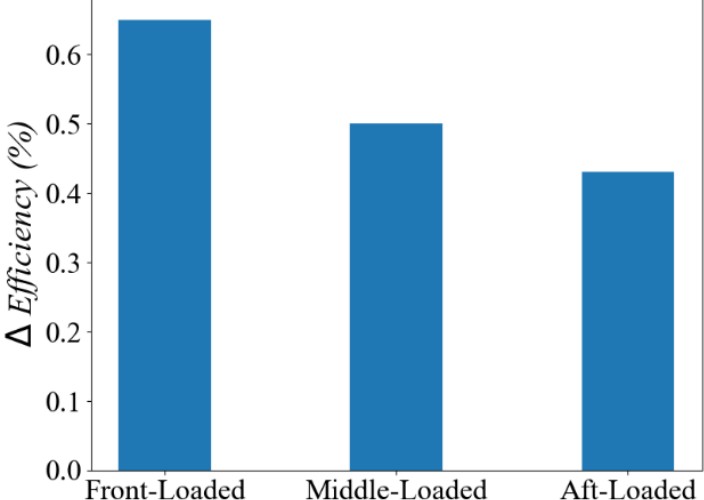

**Figure 18.** Efficiency improvement with three loading distributions, at design speed and design pressure ratio.

The mass flow exiting the tip gap is generated by two mechanisms: 1. directly from the leading edge (demonstrated as green streamlines, as shown by 'M' in Figure 19), and 2. from the pressure side gap (demonstrated by red streamlines, as shown by 'N' in Figure 19). The first part is largely sensitive to the flow attack angle and the low-momentum fluid near the casing endwall, while the second part is mainly driven by the pressure difference between the pressure side and the suction side across the tip gap.

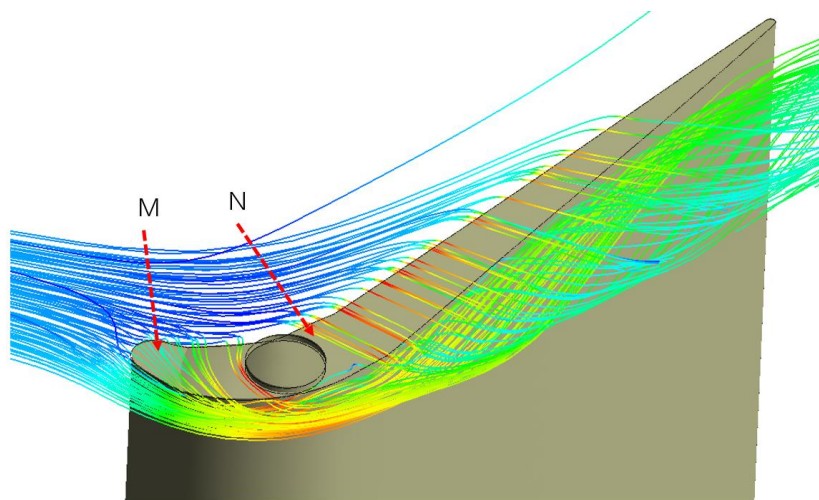

**Figure 19.** Streamlines near the tip clearance.

Corresponding to the leakage-forming mechanisms, there are two reasons why the front-loaded design tends to obtain a higher efficiency improvement; the first reason is that the most aggressive pressure gradient across the tip surface is more concentrated within a smaller region, compared to the middle-loaded and aft-loaded cases, according to the loading distributions in Figure 15b,c. As a result, the spherical convex plat, as well as the pivoting shaft, plays a significant role in blocking the leakage mass flow. The second reason is that, in the front-loaded case, the convex blockage is imposed close to the leading edge. The leaking mass flow from the leading edge part is thus blocked by the plat, while in the middle-loaded and aft-loaded cases, the blockage effect due to the leading edge leakage flow is neglected. This indicates that from the perspective of aerodynamic design, the front-loaded blade profile is more suitable to being applied with this novel convex plat.

## 5. Conclusions

To design a better variable geometry turbine, a novel spherical convex plat with a pivot shaft is proposed with numerical methods. Mesh independence is conducted to ensure the credibility of numerical skills. Based on an understanding of the loss mechanism, a universal design philosophy is proposed for different loading blade profiles. With the application of the new structure, better aerodynamic performance is achieved. The main conclusions are as follows:

1.  The installation position of the spherical convex plat with a pivot shaft should be carefully chosen based on the tip/hub surface pressure distribution with a uniform clearance. The most aggressive pressure gradient corresponds to the maximum leakage region, which is the optimal choice for the spherical convex plat. Within the range considered, an evident improvement of 0.4–3.0% is achieved, depending on the working conditions.
2.  A radially restacked vane is investigated with the novel convex plat. The hub profile is slightly moved towards the leading edge. As a result, the maximum leakage region on the hub surface is radially overlapped with the maximum leakage location on the tip surface. This restacked design obtains another 0.2% efficiency improvement, which emphasizes the design philosophy.
3.  Three typical loading profiles, i.e., the front-loaded, the middle-loaded, and the aft-loaded blades, are artificially designed. The results show that the front-loaded profile design has more potential to improve the convex design. This is contributed to by two mechanisms: the blockage of most aggressive pressure gradient flow across the tip surface and the reduction in the leakage flow from the leading edge part. From the perspective of aerodynamic design, a front-loaded choice is more suitable for better efficiency.

**Author Contributions:** Conceptualization, K.Z.; methodology, K.Z.; software, K.Z.; validation, K.Z.; formal analysis, K.Z.; investigation, K.Z.; resources, K.Z.; data curation, K.Z.; writing—original draft preparation, K.Z.; writing—review and editing, K.Z.; visualization, K.Z.; supervision, K.Z.; project administration, K.Z. and X.Z.; funding acquisition, K.Z. and X.Z. All authors have read and agreed to the published version of the manuscript.

**Funding:** This research was funded by National Natural Science Foundation of China (NSFC), Grant No. 52006117 and the National Major Science and Technology Projects of China (Grant No. HT-J2019-I-0006-0006, HT-J2017-II-0004-0016).

**Acknowledgments:** The first author would thank Lucheng Ji for his generous help in advising the whole project. The first author would also thank my colleague Hengtao Shi for his help in blade profile construction.

**Conflicts of Interest:** The authors declare no conflict of interest.

## Nomenclature

| | |
|---|---|
| $C_p$ | Static pressure coefficient, $C_p = (P_{01} - p)/(P_{01} - p_{ex})$ |
| $C_{P0}$ | Stagnation pressure coefficient, $C_{P0} = (P_{01} - P_0)/(P_{01} - p_{ex})$ |
| $C_x$ | Axial chord |
| $P_0$ | Total pressure |
| Re | Reynolds number |
| RANS | Reynolds-averaged Navier–Stokes |
| TLV | Tip leakage vortex |
| VGT | Variable geometry turbine |
| $\tau$ | Gap size |
| d | Spherical convex geometry |
| h | Span |
| p | Static pressure |
| r | Spherical radius |
| R | Pivot shaft radius |
| x | Coordinate in the axial direction |
| y | Coordinate in the y direction |
| z | Coordinate in z direction or spanwise direction |
| **Subscript** | |
| 0 | Total pressure |
| 1 | Stator inlet surface |
| ex | Stator outlet surface |
| t | Tip |
| h | Hub |

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
