# Peer review of "A Novel Design towards Reducing Leakage Loss for Variable Geometry Turbines"

_processes, doi:10.3390/pr11010021_

Round 1
Reviewer 1 Report
1. In chapter 3.2 Flow Field Analysis, please place the static entropy contours in the meridional section (both the stator and the rotor). Such a comparison will also allow to estimate changes and their magnitude in the flow channel. In addition, I am asking you to enlarge figures 6, 11, 16 because in their current form they are illegible.
2. What is the definition of fitness used by the authors, e.g. Figure 9. Please provide a definition. Shouldn't percentage point pp. be entered in the case of fitness gain?
3. Please provide the blade-to-blade velocity or pressure contours (at least on the pitch diameter) for front-loaded, middle-loaded and aft-loaded cases with comments.
4. I also ask for a more thorough review of the literature. Examples of works on this topic:
https://doi.org/10.1016/j.energy.2009.10.026
https://doi.org/10.1016/j.ast.2022.108012
I recommend tabulating the works and presenting the novelty in the presented case.
Author Response
Thanks for your suggestions. Please check the attached PDF

Reviewer 2 Report
Good engineering work to improve the design of a VTG turbine blade.
Notes:
The inlet/outlet boundary conditions are not described for mathematical modelling.
Fig.6: The streamlines details are not visible in fig, do it as in Fig. 18.
Fig.9,10: It is not clear what is “datum” and “convex” in the legend.
Fig. 11: The hub section should be enlarged to be clear.
Author Response
Thanks for your encouragement. We carefully revised the manuscript, please check the attached file.
